# NHS Diabetes Prevention Programme in England: formative evaluation of the programme in early phase implementation

Linda Penn,[1,2,3] Angela Rodrigues,[1,2] Anna Haste,[1,2,3] Marta M Marques,[1,2] Kirsten Budig,[1,2] Kirby Sainsbury,[1,2] Ruth Bell,[1] Vera Araújo-Soares,[1,2] Martin White,[1,4] Carolyn Summerbell,[2,5] Elizabeth Goyder,[6] Alan Brennan,[6] Ashley J Adamson,[1,2,3] Falko F Sniehotta[1,2]

[1]Institute of Health & Society, Newcastle University, Newcastle upon Tyne, UK
[2]Fuse: UKCRC Centre for Translational Research in Public Health, Newcastle upon Tyne, UK
[3]Human Nutrition Research Centre, Newcastle University, Newcastle upon Tyne, UK
[4]MRC Epidemiology Unit, University of Cambridge, Cambridge Biomedical Campus, Cambridge, UK
[5]School of Applied Social Sciences, Durham University, Durham, UK
[6]School of Health & Related Research, University of Sheffield, Sheffield, UK

**Correspondence to**
Dr Linda Penn;
linda.penn@ncl.ac.uk

## ABSTRACT

**Objectives** Evaluation of the demonstrator phase and first wave roll-out of the National Health Service (NHS) Diabetes Prevention Programme (DPP) in England. To examine: (1) intervention design, provision and fidelity assessment procedures; (2) risk assessment and recruitment pathways and (3) data collection for monitoring and evaluation. To provide recommendations informing decision makers on programme quality, improvements and future evaluation.

**Design** We reviewed programme documents, mapping against the NHS DPP specification and National Institute for Health and Care Excellence (NICE) public health guideline: Type 2 diabetes (T2D) prevention in people at high risk (PH38), conducted qualitative research using individual interviews and focus group discussions with stakeholders and examined recruitment, fidelity and data collection procedures.

**Setting** Seven NHS DPP demonstrator sites and, subsequently, 27 first wave areas across England.

**Interventions** Intensive behavioural intervention with weight loss, diet and physical activity goals. The national programme specifies at least 13 sessions over 9 months, delivered face to face to groups of 15–20 adults with non-diabetic hyperglycaemia, mainly recruited from primary care and NHS Health Checks.

**Participants** Participants for qualitative research were purposively sampled to provide a spread of stakeholder experience. Documents for review were provided via the NHS DPP Management Group.

**Findings** The NHS DPP specification reflected current evidence with a clear framework for service provision. Providers, with national capacity to deliver, supplied intervention plans compliant with this framework. Stakeholders highlighted limitations in fidelity assessment and recruitment and retention challenges, especially in reach and equity, that could adversely impact on implementation. Risk assessment for first wave eligibility differed from NICE guidance.

**Conclusions** The NHS DPP provides an evidence-based behavioural intervention for prevention of T2D in adults at high risk, with capacity to deliver nationally. Framework specification allows for balance between consistency and contextual variation in intervention delivery, with session details devolved to providers. Limitations in fidelity assurance, data collection procedures and

## Strengths and limitations of this study

► Evidence-based guidelines informed a structured review of National Health Service Diabetes Prevention Programme (NHS DPP) service specification and intervention provider documents, covering the whole implementation span from raising awareness to follow-up.
► Purposive sampling ensured that a spread of stakeholder experience across four groups (local commissioners, referrers, intervention providers and service users), and from different areas of the country, was included in qualitative research interviews and interactive focus group discussions.
► We made recommendations to the NHS DPP Management Group on intervention fidelity, risk assessment and recruitment procedures to strengthen future implementation and support future definitive evaluation.
► Opportunities to maximise learning from early phase implementation of the NHS DPP were limited by the pace of roll-out set against the timescale of the evaluation.
► Quantitative outcome data analyses and reporting was beyond the scope of this study.

recruitment issues could adversely impact on intervention effectiveness and restrict evaluation.

## INTRODUCTION

'Healthier You', the National Health Service (NHS) Diabetes Prevention Programme (DPP) offers adults in England at high risk of type 2 diabetes (T2D) an evidence-based behavioural intervention to prevent or delay T2D onset.[1] Led by a partnership of NHS England, Public Health England (PHE) and Diabetes UK, the NHS DPP is being implemented in phases, with plans for 100 000 places to be made available across England by 2020 and each year thereafter.[1] A brief description of the intervention is provided in box 1. This initiative was part of the NHS Five

> **Box 1  Description of the National Health Service (NHS) Diabetes Prevention Programme (DPP) risk assessment and intervention**
>
> The NHS DPP, an evidence-based behavioural intervention to prevent or delay the onset of type 2 diabetes (T2D) in people at high risk, is being made available to adults in England, aged over 18 years with non-diabetic hyperglycaemia (NDH), who are mainly identified through primary care and NHS health checks. The programme is commissioned and funded nationally and implemented by national and regional teams. In the first wave of implementation, recruitment includes from existing NHS records of people with NDH. Eligibility is based on glycoslyated haemoglobin A1c (HbA1c) value 42–47 mmol/mol (6.00%–6.49%) or a fasting plasma glucose FPG value 5.5–6.9 mmol/L within the 12 months prior to referral. In the first wave of implementation, each of the four national provider organisations will also recruit from community settings in one of their allocated intervention sites. These community-based recruitment pilots are being evaluated. The NHS DPP evidence-based behavioural intervention is specified in accordance with a national framework, with core goals of weight loss, improved diet and increased physical activity, and with the use of behaviour change techniques in intervention delivery. The specification and provider contracts require the intervention to be delivered face to face to groups of 15–20 adults with NDH over at least 13 sessions (totalling 16 hours) with a minimum of 9 months' duration. The initial provider contract terms are for 2 years before contract renewal, which will provide the first opportunity for any major variation to the programme.

Year Forward View,[2] which emphasised the importance of prevention and public health to the sustainability of the NHS and economic prosperity of Britain.

Tackling the increase in T2D is vital to the sustainable future of the NHS.[3] In 2015, 3.8 million people in England aged over 16 years had diabetes.[3] Prevalence of diabetes is greater in areas of socioeconomic deprivation,[4] and people from south Asian and black ethnic groups are twice as likely to have diabetes compared with people from white or other ethnic groups (15.2% vs 8.0%, respectively).[3] About 90% of people with diabetes have T2D, which is linked to obesity and largely preventable. Costs to the NHS for T2D treatment are currently £8 billion each year, and T2D is increasingly affecting younger and working age people.[5]

Effectiveness of complex behavioural interventions, often referred to as lifestyle interventions, to prevent or delay T2D onset in people at high risk was first demonstrated in randomised controlled trials (RCTs) in Finland[6] and the USA.[7] These RCTs were conducted in adults with impaired glucose tolerance (IGT) (ie, blood glucose values between 7.8 mmol/L and 11.1 mmol/L 2 hours after a standard 75 g oral glucose tolerance test[8]). The first T2D prevention RCT in England was based on the Finnish Diabetes Prevention Study (DPS) protocol and demonstrated a similar T2D risk reduction of 55% in the intervention compared with the control group.[9] Subsequently, studies in different populations have shown the beneficial effect of lifestyle intervention in reducing T2D onset in adults at high risk, including translational programmes delivered in primary care or community settings, some of which were implemented on a large scale.[10] However, the risk reduction in translational programmes was generally less than in the early trials. Pooled T2D risk reduction of 26% in those receiving an intervention compared with usual care was reported in a recent review of pragmatic lifestyle interventions for diabetes prevention in UK routine practice commissioned by PHE to inform the specification of the English NHS DPP.[11]

## Demonstrator site phase and first wave implementation of the NHS DPP

In 2015, seven 'demonstrator sites' from across England were selected to provide a variety of T2D prevention programme models and populations, offering examples of intervention service delivery to inform subsequent NHS DPP development and roll-out. The first wave of the NHS DPP was then commissioned nationally, to be implemented by a national and regional team, and delivered by four provider organisations that had capacity to deliver the intervention across England. First wave NHS DPP roll-out started in May 2016 in 10 of the first wave areas, where referrals to the programme began between June and September 2016. In June 2016, roll-out continued, to include the remaining 17 first wave areas where referrals to the programme began between August and November 2016. First wave areas were invited to participate in mini-competitions that set out local context and related needs to inform local variations in intervention provision. The mandate for the first wave referred to generation of at least 10 000 referrals, with up to 20 000 NHS DPP places to be made available in 27 areas, across England.

Our independent formative evaluation, of the demonstrator site phase and first wave implementation of the NHS DPP, was commissioned by the UK Department of Health to inform subsequent NHS DPP implementation and evaluation. We examined: (1) intervention design and provision, in relation to the evidence base, and procedures to assess intervention fidelity; (2) risk assessment procedures and recruitment pathways; and (3) data collection, monitoring and evaluation. We provided detailed and explicit recommendations, based on findings from the formative evaluation, to inform the national programme management group and other decision makers on programme quality, improvements and future implementation and evaluation.

## METHODS
### Framework for evaluation
In planning the evaluation, we drew on the Medical Research Council (MRC) guidance for development and evaluation of complex interventions to improve health[12] and the MRC guidance for process evaluation of complex interventions to improve health.[13] Evaluation methods included structured document review, comparing the programme with the recommendations in National Institute for Health and Care Excellence (NICE) PH38,[10] and

## Box 2  Social factors, service specification and fidelity assurance: qualitative research themes and illustrative quotes

### Social factors

*Benefit of group support*: 'It was just [being] in a group and that was good…we've made friends with each other. I think it helps'. (Interview, service user 18, female, age 76 years)

'We're all in the same boat, yes, it's lovely. I'll miss them when it's finished'. (Interview, service user 4, female, age 52 years)

*Challenge of socialising and behaviour change*: 'When you go to somebody's home and they've invited you in and they've prepared a meal for you, it's very difficult to say, "I won't eat that. I can't eat that. I shouldn't eat that." ' (Interview, service user 3, female, age 61 years)

*Influence on others:* 'Yes, I also got my brother into it you know, my older brother he started coming to the gym with me. Plus when this is over my wife is coming to the gym with me'. (Interview, service user 16, male, age 67 years)

'And they're spreading the message… We know they go away, not only just talking to their family, but they take the ideas to the workplace, to their friends' . (Focus group, workshop 1)

'So their lives will be better, but also, maybe their friends and family might get a few tips, and they might be better'. (Interview, intervention deliverer 7, male, age 44 years)

### Service specification

*Flexibility and use of intervention scripts*: 'The way the conversation goes with the patient determines, a lot of the time, which script you would base your (information) on, because quite often you'll find the conversation will bring out something they're not sure of'. (Interview, intervention deliverer 15, female, age 58 years)

*Tailoring and adaptation*: 'You can use the national model and tailor it to the local one and look at the ethnicity and devise the programme accordingly to fit your community'. (Focus group, workshop 1)

'You have to adapt the course for a whole range of intellectual abilities,… And different cultural understandings'. (Interview, local commissioner 1, female, age 57 years)

*Progressive variation:* 'As people start to work through the backlog and people understand slightly more about what works and what doesn't work there can be more variation'. (Focus group, workshop 2)

### Fidelity assurance procedures

*Staff qualities and training*: 'They don't have to be diabetes specialists, but I think they need to be specialist trained in order to actually deliver a high level of intervention programme … you don't want the confidence and authority without knowledge'. (Focus group, workshop 1)

'I think [the deliverer] is a very likeable person, easy to listen to and quite clearly was quite keen on her subject'. (Interview, service user 7, male, age 71 years)

'I trained as a nurse. I've also done my level two gym instructor, level three GP exercise referral, currently studying towards level 4; dealing with diabetic and obese clients'. (Interview, intervention deliverer 12, female, age 30 years)

'It's behaviour change, motivational interviewing, brief advice, brief intervention and all those kinds of behaviour change qualifications and experience'. (Interview, intervention deliverer 1, male age 35 years)

*Mentoring and refresher training:* 'I watched how other people deliver it, just to give myself a little bit more confidence, really'. (Interview, intervention deliverer 8, female, age 51 years)

'I think refresher training is always good. Things are always changing, new research is coming out, there are new guidelines. So refreshing is

*Continued*

## Box 2  Continued

important just to keep up to date'. (Interview, intervention deliverer 12, female, age 30 years)

qualitative research involving individual interviews and interactive focus group discussions with key stakeholders, including service users.[14] In this paper, we report brief details of methods and key findings.

### Research governance and ethics

All interview and focus group participants gave written informed consent to take part. Research governance and reporting strategies were agreed between the evaluation team and the UK Department of Health, Research Development Directorate (which commissioned the evaluation) and the National Institute for Health Research (NIHR) School for Public Health Research (which funded the research).

### Methods to address objectives

#### Document review

We reviewed and appraised programme documentation by systematically mapping written information supplied by providers (using demonstrator site applications and first wave provider procurement prospectuses in separate reviews) and the NHS DPP service specification, against recommendations in NICE guidance PH38,[10] equality indicators,[15 16] fidelity,[17] data collection and quality assurance procedures. We defined fidelity to mean 'intervention delivered as intended' thus encompassing components of adherence to specification and competence to deliver the intervention.[17]

#### Patient involvement

Patients were involved in review of intervention materials designed for service users and the service user questionnaires that were drafted as part of this research.

#### Qualitative interviews and interactive focus-group workshop discussions

We used individual semistructured interviews (n=62), conducted mostly by telephone, and interactive focus group discussions, delivered through two full day workshops each with three by three parallel 1 hour sessions (total 18 hours), to explore stakeholder experiences.[18] Purposive sampling was used to obtain a spread of experience across four stakeholder groups: local commissioners, referrers (mostly healthcare professionals), intervention providers and deliverers, and service users (all service users were interviewed in the demonstrator site phase). Topics addressed included intervention provision (participant experience, access and equity, fidelity of intervention delivery and staff training), referral and recruitment pathways, data collection and sustainability of behaviour change.[13 19] We used the NHS DPP logic model diagram, which detailed resources, actions and outcomes, as a visual aid in focus group discussions. All focus groups

and interviews were digitally audio recorded, transcribed and analysed using the Framework method.[18] Early findings were used to inform later interviews and focus group discussions, with the aim of achieving thematic saturation. Qualitative data were collected, coded and checked by AR, AH, LP, KB and MMM with use of NVivo V.10 to facilitate data management.[20]

### Data collection for monitoring and evaluation

To inform subsequent evaluation of NHS DPP implementation and equity, we explored data collection and fidelity assessment procedures by comparing data collected in demonstrator site programmes and data in the NHS DPP 'minimum data set' (ie, the list of data items that each national provider organisation in the first wave NHS DPP implementation was contractually committed to collect) with data requirements detailed in NICE PH38.[10] We used qualitative research findings to develop tools for use in intervention feasibility, acceptability and fidelity assessment, and we invited stakeholder feedback to refine these tools. We worked with a key provider contact to coproduce a flow chart of recruitment pathway data and examined T2D risk factors using risk scores validated in a UK population (QDiabetes,[21] Cambridge Risk Score,[22] the Finnish Diabetes Risk Score[23] and the Diabetes-UK risk score).[10]

### FINDINGS

The NHS DPP Management reported that: (1) from 1 April 2015 to 31 March 2016: 6577 people at high risk were referred to the seven demonstrator site programmes (94% of the target) and of these 3165 people (48% of those referred) started the intervention; (2) from 1 April 2016 to 31 March 2017: 3170 people were referred to the four ongoing demonstrator site programmes and of these 2220 people (71% of those referred) started the intervention; and (3) from 1 April 2016 to 31 March 2017: 43 606 people with NDH were referred to first wave programmes (116% of the target). Given the expected

time lag between referral intervention uptake and data monitoring, the management group reported that 49% of those referred up to 31st January 2017 had attended the initial programme assessment by 31 March 2017.[24] The management group also reported that financial support had been given to local health economies to facilitate demonstrator site programmes and first wave implementation of the NHS DPP.

### Intervention provision, in relation to the evidence base, and procedures to assess fidelity of intervention delivery

We noted discrepancies between the detailed guidance on behaviour change techniques (BCTs) in NICE PH38[10] and the sparse information on BCTs in the draft NHS DPP service specification that was supplied to us during the demonstrator phase. However, the NHS DPP service specification was updated for the first wave to include more detail on BCTs and a theory of change[13] (logic model). First wave provider documents linked standard BCTs with their expected mechanisms of action, BCTs for sustained behaviour change were often included and all providers detailed staff training on behaviour change strategies and use of theory-driven BCTs. Additional detailed description of intervention component mapping are provided as appendix A (online supplementary file 1). Intervention fidelity procedures, including staff qualifications and training, were assigned as provider responsibilities. All first wave provider documents set out dietary and physical activity advice that complied with the NHS DPP service specification (as was required in the procurement process) and included detailed session plans with information about weight loss and regular weighing of participants. Intervention duration and intensity also complied with the service specification, although session distribution differed between providers, as shown in table 1.

Documented variation between first wave providers on additional services included free access to local exercise facilities, signposting to NHS choices for weight loss[25] and free signposting to Slimming World online.[26] Provision

**Table 1** First wave intervention delivery approach across providers

| | National providers | | | |
| --- | --- | --- | --- | --- |
| | **A** | **B** | **C** | **D** |
| Core sessions | Six group closed core sessions (of 90 min), held on alternate weeks | Six group sessions, held weekly | Four group core sessions (of 90 min), held weekly | One (individual session) and seven group sessions (of 60–90 min), held weekly |
| Follow-up/ maintenance sessions | One individual review session (of 60 min) held at month 3 Four open group maintenance sessions (of 60 min) | Four sessions, held fortnightly Six sessions held monthly | Nine maintenance sessions, held monthly | Six programmed support contacts over the 12 months of the Diabetes Prevention Programme, held at months 2, 3, 4, 5, 6, 9 and 12 |
| Review sessions | Follow-up reviews, held at 6 and 9 months | Two review sessions | | |

**Table 2** Qualitative research participants

| Stakeholder group | Individual interviews | Focus groups (WS1) | Focus groups (WS2) |
|---|---|---|---|
| Local commissioners | n=13 (6 m, 7 f) 26–49 min | n=6 (3 m, 3 f) | n=21 (6 m, 14 f) |
| Referrers/healthcare professionals | n=10 (5 m, 5 f) 23–45 min | n=2 (1 m, 1 f) | |
| Intervention deliverers (demonstrator phase) | n=15 (4 m, 11 f) 21–65 min | n=6 (1 m, 5 f) | |
| First wave provider key contacts | n=3 (1 m, 2 f) 60 min | | n=3 (1 m, 2 f) |
| Service users (demonstrator phase) | n=20 (9 m, 11 f) 23–39 min | n=1 (1 f) | – |

f, female; m, male; WS, Workshops at Newcastle University London campus: WS1: (23 March 2016): WS2- (12 December 2016).

of single sex groups and accessible venues, to address equality issues, were set out in provider plans. However, it remained unclear how realisation of different adaptations to the programme will be triggered or how they will be prioritised within a proportionate universalism framework (as outlined in the Marmot Review).[27] The wide range of possible adaptations, presented by providers, risks accountability in the absence of explicit plans to monitor the impact of specific adaptations. Document review identified that fidelity procedures, including staff qualifications and training, were assigned as provider responsibilities. There were limited contractually agreed fidelity procedures that were applied in common across all providers.

A summary of qualitative research participant interviews and focus groups is provided in table 2.

Themes identified through qualitative research of stakeholder perspectives included: social factors, such as group support, socialising and influence of service users on other people; service specification, such as flexibility and use of intervention scripts, tailoring, adaptation and progressive variation in intervention provision; and fidelity procedures, including staff qualities and training, mentoring and refresher training. Service users benefited from the peer support of their DPP group and some spoke about social support from family and friends. They also highlighted some challenges and opportunities in the social nature of the DPP target behaviours. Intervention delivery staff spoke about flexibility in their delivery of the intervention and how this might change with context and experience. Staff qualities and fidelity procedures were also discussed, with consensus on the importance of these. Illustrative quotes for these themes are provided in box 2.

Concerns about sustainability, post-NHS DPP intervention provision and the importance of linkage with existing services were important themes. The opportunity to make use of the NHS DPP to promote health messages was highlighted. Illustrative quotes for these themes are provided in box 3.

### Risk assessment procedures and recruitment pathways
Document review showed that referral was via primary care or NHS Health Checks[28] in all demonstrator sites and some also reported community-based recruitment, but allocation of recruitment responsibilities lacked clarity.

Risk communication[29 30] procedures were reported in one demonstrator site only. In the NHS DPP first wave, agreements with local health economies covered referral targets, with recruitment being clarified as a provider responsibility, hence partnership working was required. In the demonstrator site phase, community-based awareness raising in black and minority ethnic communities was included in baseline documentation. Community-based recruitment was contracted in four of the first wave areas (one for each provider) in order to consider additional future recruitment approaches for harder to reach groups.

**Box 3    Sustainability, postintervention provision, linkage with existing services and opportunity to promote health messages: qualitative research themes and illustrative quotes**

*Sustainability, postintervention provision:* 'There are enormous funding pressures on so many of our services that to argue for investment in prevention is quite a difficult ask'. (Interview, local commissioner 7, female, age 60 years)

'Yes, and if it's not worked, there should be something else, because that's what the whole point is isn't it, it's to try and stop this diabetes epidemic? To try to reduce the amount of NHS time that will need to be spent on people who become diabetic and the problems that that causes'. (Interview, service user 19, female, age 66 years)

'I feel healthier, I feel more positive and I feel very determined not to let this thing happen to me if I can do anything about it'. (Interview, service user 3, female, age 61 years)

*Linkage with existing services:* 'We don't have a lot of money to commission these things, we're just looking at the best ways of working, so we're linking it to other things as a borough'. (Focus group, workshop 2)

'It's trying to bring your local services along that same journey, isn't it so they're working together right from the very start of the programme'. (Focus group, workshop 2)

*Opportunity to promote health messages:* 'It's an additional resource in the community that people are more networked and actually link in with each other'. (Focus group, workshop 2)

'Maybe [add] conversations about how they could be sharing their knowledge with their family- are they improving their family meals'. (Focus group, workshop 1)

'Using the Diabetes Prevention Programme as a launch pad for a wider public debate on food, sugar tax, population level risk'. (Local commissioner 4, male, age 63 years)

**Box 4    Risk assessment procedures and recruitment pathways: qualitative research themes and illustrative quotes**

**Crucial role of primary care in recruitment, allocation of supportive resource and effective risk communication:**

'The ones [GP practices] with incentives have got started much quicker'. (Focus group workshop 2)

'The real driver for that response rate has been the fact that letters have been from GP practices'. (Focus group workshop 2)

'Push the idea that it is in the GPs financial interest also to get involved'. (Focus group workshop 2)

'Because sometimes, the practice managers can be the gate keepers and … the practice manager says [to a referrer] "No, you've got to do your influenza clinic" '. (Focus group workshop 1)

'If you've only got one side driving it's going to be a struggle'. (Focus group workshop 2)

'Health professionals across the board should be having much more information in their minds so they can have that brief intervention, that brief every contact counting'. (Focus group workshop 1)

' For me, one of the main elements is not the eligibility… It's the communication of risk. In terms of a conversion rate I think that's really, really important'. (Focus group workshop 2)

**Contextual challenges, equality and community-based recruitment:**

'We have a huge geographical area… with a very dispersed population and we're also very remote. We don't have providers knocking on our door because it [intervention delivery across a wide rural area] is not viable'. (Focus group workshop 2)

'We are getting what they call the low hanging fruit to start with. The longer it goes on the harder it is going to be to identify people and to get them to engage'. (Interview, local commissioner 3, female, age 58 years)

'If we really want to target the clients that we really want to engage with, it's about…working with those community leaders'. (Focus group workshop 1)

'We're using community based settings all over…we try to make sure they're on a bus route'. (Interview, intervention deliverer 14, female age 52 years)

**Local commissioners, shared learning and allocation of roles and responsibilities:**

'I would have a plea as to a repository of information, so that we can look at different letter templates that have been successful'. (Focus group workshop 2)

'I have come along today and listened to people with real experience in it… lessons learnt, different uptakes… none of that's shared and that sort of sharing would save us a load of work'. (Focus group workshop 2)

'We are going to change our referral pathway and get practices to phone patients because we have learnt from other areas there's a much higher conversion rate'. (Focus group workshop 2)

'What was the role of NHS England? What was the role of the local partnership? What was the role of the provider? That's a whole roles and responsibility difficulty around the tripartite arrangement. For me if something goes wrong, I'd be thinking, "Well, how do you deal with it?" '. (Focus group workshop 2)

Qualitative research highlighted respondents concerns and challenges around referral and uptake including: the crucial role of primary care in recruitment, allocation of

resources and training to support this role and effective risk communication; contextual factors, equality issues and the need for community based recruitment; and the difficulties of intervention provision in sparsely populated rural areas. Local commissioners flagged the benefit of shared learning to maximise efficiency in the referral pathway and the difficulties in understanding the allocation of roles and responsibilities. Illustrative quotes in support of these qualitative research themes are provided in box 4.

In the demonstrator site phase risk score use and blood testing were documented, with the Diabetes UK risk score[10] and HbA1c blood test being the usual measures. The service specification for the NHS DPP first wave relied exclusively on assessment of NDH as the criterion to determine high risk for intervention eligibility, and risk scores were not used.

### Data collection to support monitoring and evaluation

Demonstrator site data collection was usually documented in terms of primary care data systems. First wave providers were contracted to collect a 'minimum data set' of items. However, there were uncertainties regarding data collection procedures and measurement protocols, with potential for variation between providers. We noted variation between providers in collection of data additional to items in the minimum data set (such as dietary data). First wave providers were required to notify general practitioners when people were discharged from the NHS DPP and to liaise locally to ensure these data could be integrated in clinical systems. However, in focus groups with local commissioners, reporting procedures to inform timely local public health monitoring of the NHS DPP was raised, and the need for better data sharing and timely feedback to monitor local intervention progress, in relation to referral targets and process outcomes, was a strong theme in focus group discussions. Timescales in relation to learning from the demonstrator sites were also highlighted, with the pace of roll-out being viewed as an important limitation and missed opportunity.

Documentary information on data collection to monitor recruitment routes and intervention uptake was limited. This limitation was also identified through qualitative research on recruitment in relation to subgroups (eg, gender and ethnicity) and type of contact (eg, mail-drop, face to face or phone). However, the need to balance data collection with respondent burden to avoid detrimental impact on participant engagement, especially where there might be language or literacy issues, was highlighted.

We identified risk assessment and referral costs as contributors to the overall costs and efficiency of NHS DPP implementation. Attribution of these costs (eg, to primary care) and how they might be included in economic analysis of the NHS DPP in later phases was unclear. Comparison of the NHS DPP minimum data set with incident risk score data is detailed in table 2. In qualitative research local commissioners and intervention

---

**Box 5  Monitoring, evaluation and data collection: qualitative research themes and illustrative quotes**

**Data sharing and timely feedback to monitor local intervention progress and learn from the demonstrator sites**

'NHS England have commissioned a CSU (commissioning support unit) to do a lot of data reporting, and extraction from the providers. So I'll be expecting to have regular reporting from them. I think the intention, as well, is to have relatively regular meetings with the provider themselves. With the coding that we have on [GP data base]we should be able to see on there certainly the number of people who've been referred, and then there should be coding going back into [GP data] I would hope as well. So we should be able to see the number of people who've completed.

'In terms of the quality of the actual intervention itself, I think we will be probably relying on patient questionnaires, and other qualitative feedback that's provided. I think it's within the service specification, and data reporting, that they need to do that. So we'll probably rely on that I think'. (Interview, local commissioner 2, male, age 33 years)

'You would look at the outcomes of the service that was being provided and saying, "If that is not achieving the long-term changes then do we need to look at changing it or trying something different? That is the same I think with most commissioning, if the service you are commissioning isn't providing the outcomes you need it to achieve then you have to look again" '. (Interview, local commissioner 3, male age 58 years)

'I don't think the timescales have been good at all. I'm very disappointed that the demonstrator sites haven't been demonstrator sites at all, because we're barely up and running and the national provider's almost been procured. I don't know where all the learning is going. I think if the pilot programmes had been allowed a little bit more time to set up and to run, and then for them to conclude, and then take the learning from that to fashion, nationally, what it is that's going to be provided then I think that would have been a lot better, but I think this short-termism has not enabled people to really get to grips with learning'. (Interview, local commissioner 1, female, age 57 years)

**Data to monitor recruitment routes and intervention uptake**

'If we wanted to do some local health equity audit and look at whether the right people were getting into the right programmes, would we actually get the level of data that would enable us to do that? I think that at that stage NHS England said they didn't know. As far as I am aware that is probably still the case'. (Focus group workshop 2)

**HbA1c testing and values**

'And out of 307 [number referred] 10 have been ineligible for the programme after they've gone through the test, because the provider will test them again, their HbA1c'. (Focus group workshop 2)

'I mean there is the group who have just slipped into diabetes. That is a tricky area as well because they can't be offered the programme'. (Focus group workshop 2)

'The problem is when you get a different result from a point-of-care (HbA1c) testing from the venous testing that's gone on beforehand through a GP, or other parts of primary care'. (Focus group workshop 2)

---

providers expressed concerns about HbA1c testing and eligibility including: what to do when test values were just over the upper limit, when there were differences in repeated tests and when there were discrepancies between point of care values compared with standard venous blood test values, which could adversely impact outcome evaluation[31][32] (iii)). New guidance on eligibility criteria and HbA1c measures was issued in early 2017 by the NHS DPP implementation team, following a consultative exercise, to address eligibility issues. Illustrative quotes in support of these qualitative research themes are provided in box 5.

## DISCUSSION

### Principal findings

The NHS DPP first wave specification reflected the evidence base[10] and provided a service framework with sufficient flexibility to support balance in intervention delivery between consistency and limited variation to accommodate local contextual and cultural adaptations.[33] Providers, with capacity to deliver NHS DPP compliant interventions on a national scale, were commissioned to deliver the intervention in 27 first wave areas. Responsibility for detailed session planning was devolved to intervention providers. Procedures to ensure fidelity of intervention delivery lacked clarity and were reliant on provider driven (internal verification) procedures with[34] inevitable variation between providers. How variations in intervention provision (eg, to accommodate cultural adaptations) within each provider organisation were prioritised, triggered, actioned and monitored[27] were unclear and this omission is likely to compound uncertainties regarding fidelity of intervention delivery and adversely impact on effectiveness, evaluation and monitoring for service improvement.

First wave NHS DPP risk assessment procedures differed from NICE guidance.[10] Recruitment of people with an existing primary care NDH record might be considered a reasonable and pragmatic first wave strategy, because these people are known to be at risk of T2D. However, modification will be needed in subsequent phases to improve consistency and equity in relation to risk stratification. Concerns regarding reliability of point of care HbA1c testing will affect outcome evaluation.[31][32] Distribution of stakeholder responsibilities and resources to support recruitment, staff training, linkage to other services, sustainability of the programme and behaviour change maintenance require clarification to maximise likely effectiveness of the programme going forward.

Collection of a common 'minimum data set' by providers supported basic monitoring, but additions to these data, such as including quality of life assessment[35] (now added) and data on routes to recruitment, risk factors and participant behaviours, will be needed to support evaluation and programme improvement.

### Strengths and limitations of the evaluation

We applied a systematic approach, in both demonstrator site and first wave evaluations, to appraise the baseline programme documents. We appraised elements across

the entire programme of activities from awareness raising to follow-up, avoiding a narrow focus on intervention delivery. We used recommendations in NICE guidance[10] to structure the review and appraisal of service specification and provider documents, thus assessing the service specification in relation to the evidence base: an approach that might be transferable to other intervention programmes. The decision to start with a demonstrator site phase was an innovative and potentially influential initiative. However, the pace at which the first wave of the national programme was rolled out, set against the time needed to conduct meaningful appraisal, made it difficult to maximise 'learning from the demonstrator sites' to inform the first wave NHS DPP, although this was the stated aim of the demonstrator site phase[1] and of our evaluation. We explored procedures for assessment of fidelity in intervention delivery[17] and worked towards the development of tools suitable for common assessment of feasibility, acceptability and fidelity that could be used across different providers within the programme.

A strength of our qualitative research was the inclusion of four stakeholder groups (local commissioners, healthcare professionals, intervention delivery staff and service users). Conducting interviews by phone maximised resource use and delivering focus groups in a workshop setting allowed several interactive discussions to take place within the time available for the project, but restricted the number of participants. The focus of our evaluation, at this early stage of programme roll-out, was on the broad issues of national programme specification and implementation strategies. Detailed exploration of site specific organisational strategies, and variations in these, that aimed to facilitate uptake of the programme in different localities will be of increasing interest as the programme becomes embedded across England. In individual interviews and interactive focus group discussions, stakeholders highlighted the importance of primary care in programme mobilisation. As the programme is rolled out across England, and it becomes embedded in healthcare, evaluation of local organisational strategies and their effect on implementation will be of increasing interest.

Evaluation findings were used to formulate recommendations in summaries of detailed reports that were supplied to the NHS DPP Management Group at prespecified intervals, as part of the commissioned evaluation requirements. The Management Group (intervention commissioners) categorised responses to these recommendations as: A) Recommendations implemented before the evaluation report was received; B) Recommendations implemented in response to the evaluation; C) Recommendations that might be considered for the next round of procurement; D) Recommendations deferred. Provision of recommendations, based on findings, was in line with the MRC (2008) framework for the implementation phase in development and evaluation of complex interventions to improve health.[12] Liaison with the Management Group with regard to

these recommendations, including dissemination through teleconference presentation and discussion, and of collation of tables to formalise their response to recommendations, was a strength of our formative evaluation that may guide subsequent implementation phases and facilitate the assessment of impact in implementation evaluation.

### Strengths and limitations in relation to other studies

There is strong trial evidence for the effectiveness of behavioural intervention, with content and goals similar to those of the NHS DPP intervention, to reduce the risk of T2D in people at high risk.[10] Thus, the main challenge for the NHS DPP is in its implementation, especially with the ambitious scale of the project, with expected 100 000 places to be made available each year from 2020 and pace of roll-out.[36] To support implementation, there is a need to consider local contextual evidence.[37] Although the demonstrator site phase was primed to do this, the pace of roll-out restricted the utility of this phase.

The first wave NHS DPP was commissioned nationally to be delivered by four provider organisations. Detailed evaluation of local implementation of this model is of interest but was not the focus of our early phase evaluation. Other implementation models have been employed notably in Finland,[38] the USA[36] and Germany.[39] The NHS DPP is specified according to a service framework, which devolves responsibilities that allow intervention providers some discretion on the balance between facilitator autonomy, which may maintain motivation,[40 41] and structured session plans, which may support consistency in intervention delivery. This framework specification model leads to inevitable variation between intervention providers in their approach. In the NHS DPP, variations between providers are controlled by the detailed service specification and number (four) of national providers. It is unclear how the balance between consistency and flexibility in intervention provision impacts effectiveness.

European guidelines for T2D prevention recommend reference to behaviour change theories and techniques in the design of appropriate interventions.[42] The Good Ageing in Lahti Region (GOAL) study from Finland,[43] which succeeded the DPS, included the Health Action Process Approach (HAPA) behavioural theory.[44 45] This translational study informed the Greater Green Triangle (GGT) project in Victoria, Australia,[46] which in turn influenced more extensive diabetes prevention programmes in Australia.[47 48] Both the GOAL[43] and GGT[46] 'real world' programmes demonstrated likely effectiveness. However, a review of reviews found increased effectiveness of physical activity and dietary interventions in people at risk of T2D to be associated with a cluster of behaviour change techniques rather than a particular theory.[49] The HAPA model is included in the NHS DPP change theory, although it is not clear how it might be actioned. Listing behaviour change techniques with specific associated delivery strategies might be more effective and more informative for future evaluation.

Early diabetes prevention trial experiences suggest that their format, which included personalised individual behavioural consultation,[50] mean intervention duration of around 3 years and the resource and respondent burden of risk assessment and detailed data collection, limits the external validity and utility of these research findings for large-scale implementation.[51–53] In particular, the shift to more pragmatic risk assessment and group-based intervention delivery is likely to impact adversely on effectiveness.[54]

Diabetes incidence as the outcome of interest requires long-term follow-up. The NHS DPP stipulates a minimum 9 months of intervention, but longer follow-up is needed to properly assess diabetes incidence over time.[10] Changes to the National Diabetes Audit that are designed to facilitate longer term follow-up of NHS DPP service users and others with NDH have been implemented, thus providing support for long-term evaluation of the initiative. In other translational programmes, intermediate health and behavioural outcomes have been used for interim evaluation within a shorter timeframe.[10] Weight loss is a simple and objective candidate for interim outcome analyses, but an assumption that weight loss will equate to prevention of T2D, similar to that achieved in trials, in differently identified high-risk populations is, flawed.[55] It seems probable, for example, that the physiological differences between a population identified by IGT and a population identified by IFG (or raised HbA1c) will affect their overall capacity to benefit from lifestyle intervention and thus impact on its effectiveness.[56] Although the incomplete overlap between different NDH categories was illustrated in a recent review, the implications of these different NDH categories for intervention effectiveness were not drawn out in this paper.[57]

### Meaning of the study: possible explanations and implications for clinicians and policymakers

The decision by the NHS DPP Management Group to first focus on people already identified with NDH through routine clinical practice is understandable, because these people are already known to be at risk of T2D. Risk assessment as advocated in NICE guidance (PH38) was a two-stage process.[10] In suggesting this two-stage approach, the NICE programme development group (PDG) acknowledged the scarcity of evidence in this area and included research recommendations identification and monitoring (1). Over time, the NHS DPP could provide opportunities to investigate pragmatic, efficient and equitable recruitment procedures and criteria to assess high risk and capacity to benefit from intervention to prevent T2D.

High-risk and population level approaches to T2D prevention are included in the current NHS England policy.[1 58 59] However, the engagement of people at high risk in intensive behavioural intervention may impact on others within their sphere of influence,[60 61] thus development and evaluation of a high-risk approach should take into account opportunities to use individual engagement in healthier lifestyles for wider health and social benefits.[62]

Our experience reflects the more general challenges inherent in conducting external evaluation of programmes that are continually evolving. We sought to improve the utility of this formative evaluation by supplying recommendations based on findings to commissioners, and we were able to elicit their formal response to each recommendation.[13]

### Unanswered questions and future research

A formal outcome evaluation for effectiveness and cost-effectiveness of the NHS DPP implementation has been commissioned by the NIHR.[63] As part of the formative evaluation, we drafted tools to support feasibility, acceptability and fidelity assessment of the NHS DPP intervention and improve data collection on recruitment routes to inform equity of the service provision.

Capacity to benefit from lifestyle intervention may be affected by multiple factors, and we welcome the opportunity that the national scale NHS DPP, if supported by improved fidelity and data procedures, could afford to push forward empirical research in this respect.

As a standard 'usual care' comparator, the NHS DPP nationally commissioned service has potential to support further research on lifestyle intervention for prevention of T2D, which could not otherwise be achieved, given the diversity in implementation and evaluation inherent in locally commissioned programmes.

### CONCLUSION

When fully implemented the NHS DPP will provide an evidence-based lifestyle intervention for prevention of T2D in adults at high risk, with provider capacity to deliver the intervention on a national scale. Formative evaluation of first wave NHS DPP implementation found that the intervention specification reflected current evidence, while allowing balance between consistency and contextual variation in intervention delivery, with detailed session planning devolved to providers.

Limitations in intervention fidelity assessment and data collection procedures are likely to impact adversely on intervention effectiveness. Improvements are needed to ensure the intervention is delivered as intended with support for evaluation and programme improvement.

Inviting people already identified as at risk (with NDH record) to participate in the NHS DPP may be an appropriate first wave strategy. However, compliance with NICE guidance for two-stage assessment of high risk will be needed to improve consistency, equity and cost-effectiveness for future implementation phases.

It is unlikely that one intervention model alone will engage all those at high risk or be equally effective for all participants. The national scale and detailed framework specification of the NHS DPP, if supported by improved fidelity assurance and data collection procedures, could

afford opportunities to design and test variations to the standard intervention to maximise its impact. To facilitate such empirical research, clear differentiation between planned and unplanned variations within the national programme is needed, emphasising the need for improved fidelity assessment and data collection procedures.

**Contributors** FFS, AJA, MW, CS, EG, AB, RB and LP designed the evaluation and secured funding for the study. AH, AR, MMM, KB, FFS, ABA and LP conducted fieldwork, and KS and VA-S assisted with data analysis. All authors contributed to interpretation of analyses, study reports and drafts of this summary paper. All co-authors have reviewed and agreed this final draft of the paper that is submitted for publication.

**Funding** The 'NHS Diabetes Prevention Programme: process evaluation of the demonstrator phase and consensus building for a common evaluation framework and data-set for programme implementation' and the 'Formative evaluation of the First Wave of the national implementation of the NHS Diabetes Prevention Programme' research studies were funded via the NIHR School for Public Health Research (SPHR). The work was undertaken by Fuse: a UKCRC public health research centre of excellence. Funding from the British Heart Foundation, Cancer Research UK, Economic and Social Research Council, Medical Research Council and the National Institute for health Research is gratefully acknowledged. This paper presents a summary of these two independent but linked research studies. The study sponsors had no influence on the content of this manuscript.

**Disclaimer** The views expressed are those of the author(s) and not necessarily those of the NHS, the NIHR or the Department of Health.

**Competing interests** All authors have completed the Unified Competing Interest form and declare: no support from any organisation for the submitted work other than that the fact that NIHR SPHR funded the research studies. MW and RB hold honorary consultant in public health posts with PHE. MW is programme director for the PHR programme at NIHR. Otherwise there are no financial relationships with any organisations that might have an interest in the submitted work in the previous three years and no other relationships or activities that could appear to have influenced the submitted work. The NHS Diabetes Prevention Programme, on which these independent research studies were conducted, was funded by a partnership of NHS England, Public Health England and Diabetes UK. Research governance procedures were agreed and implemented before the studies reported here commenced, to ensure the independence of our research. During the research period, we regularly reported to the NIHR SPHR and provided executive summaries, with recommendations resulting from these formative evaluations that were made available to the NHS Diabetes Prevention Programme Management Group.

**Patient consent** Not required.

**Ethics approval** We obtained NHS ethical approval and Health Research Authority research governance agreements for stakeholder interviews (IRAS number 190418) and ethical approval from Newcastle University Faculty of Medical Sciences ethics committee for stakeholder workshops (FMS number 3475/2016).

**Provenance and peer review** Not commissioned; externally peer reviewed.

**Data sharing statement** Data are available for further analysis from the corresponding author.

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
