## [Reviewer comments · BMJ Open]

ARTICLE DETAILS

TITLE (PROVISIONAL)	The NHS Diabetes Prevention Programme in England: formative evaluation of the programme in early phase implementation
AUTHORS	Penn, Linda; Rodrigues, Angela; Haste, A; Marques, Marta; Budig, Kirsten; Sainsbury, Kirby; Bell, Ruth; Araujo-Soares, Vera; White, Martin; Summerbell, Carolyn; Goyder, Elizabeth; Brennan, Alan; Adamson, Ashley; Sniehotta, Falko

VERSION 1 – REVIEW

REVIEWER	Alvaro Sánchez Pérez Primary Care Research Unit of Bizkaia-Osakidetza, SPAIN
REVIEW RETURNED	18-Sep-2017

GENERAL COMMENTS	This is a very relevant manuscript for those involved in the scale-up of proven clinical interventions. Specifically, the objective of the study is to evaluate the demonstrator phase and first wave roll-out of the NHS diabetes prevention program (DPP) in England, to inform subsequent phase. To do so, they have examined: (i) intervention design and provision, in relation to the evidence base, and procedures to assess intervention fidelity; (ii) risk assessment procedures and recruitment pathways and (iii) data collection, monitoring and evaluation. The evaluation methods included structured document review, comparing the program with the recommendations in NICE PH38, and qualitative research involving individual interviews and interactive focus group discussions with key stakeholders, including service users Due to the relevance at public health level of the general NHS Diabetes Prevention Program in England and the intended goals of the present study (informing the rule-out of future phases in a more efficient way, for both implementation and evaluation), the present manuscript has several core elements in which it might be improved. Major points First, it is not clear what and how within the MRC framework the present evaluation fits. What is the specific contribution or the specific question that the study is trying to answer within such framework? Please comment on this, especially, when the study seems to be the Phase IV translation stage. Second, and linked to the previous point, the present evaluation only considers the clinical intervention level, not informing about the most important aspect in my opinion when addressing the transfer/dissemination/scale-up of clinical interventions: the implementation strategy used to enhance the adoption and execution of the clinical intervention in the studied sites/contexts.
--

Specially, when it is pretty likely to be variable among sites and contexts, raising issues related to fidelity-flexibility, not only on how the clinical intervention has been organized and carried out, but on the implementation strategies used.

Within the different types of program evaluation (process, outcome, impact, etc.), the study actually describes factors (organized by subject) that have facilitated / prevented the 3 aspects of evaluation related to clinical intervention based on review of documentation and comparison to recommended program, and key agent opinions and perceptions extracted from interviews. As a consequence, the true contribution of the study, taking into account its strengths and weaknesses, should be more clearly specified in both the title and the body of the manuscript.

The comparison of how the programs were organized / implemented with respect to the recommendations is limited, describing in a table the operational organization of the program according to 5 sites and explaining some discrepancies in the use of BCTs and fidelity aspects. An exhaustive or at least better organized description of how the clinical intervention program has been organized / implemented at each site, and its deviation from what is recommended, would help, especially those different ways of putting it into practice. But also, it would be very interesting to know which of these procedures are successful or not, based on not only perceptions of the agents but on numbers (quantitatively). The NICE baseline measurement tool can be a possible guide on how to structure a table for that description.

Though it is a “qualitative” evaluation, and not a formal program process evaluation, further quantitative information will help to understand the magnitude of the study (ie., reach of the different screening procedures, intervention delivery procedures...by each site). This information will also help to describe which procedures and strategies are more successful.

An implementation framework or model that guides the design or at least the analysis, synthesis, and interpretation of these determinants of implementation (ie, CFIR framework) is missed. This would help not only to understand which determinants are key and why, but also to identify and / or design strategies to address such determinants in future program roll-out.

Linked to this previous point, and within the parts describing the framework and methods that have guided the evaluation, the authors refer to the mapping of the retrieved information from providers against the NICE recommendations, equality indicators, fidelity, data collection and quality assurance procedures. But none of these matched or mapped elements are clearly displayed or presented within the manuscript. They also refer to the development of some tools (page 9, last paragraph) for use in intervention feasibility, acceptability and fidelity assessment. An example of these tools, due to the help that may offer to the reading public, is highly recommendable.

Minor points

Abstract-findings: more precise or concrete results summarizing actual obtained results

A Figure Flow-diagram is highly recommendable to fully describe the structure and process (phases, sites, centers, professionals, patients referred, etc) of the study.

REVIEWER	Neal Barshes Baylor College of Medicine Houston, Texas United States of America
REVIEW RETURNED	18-Sep-2017

GENERAL COMMENTS	Page 8, paragraph 1, Introduction section: what was the goal of this study? This should be clearly stated clearly. This need not to be a hypothesis to prove or disprove, but if it is a study to ensure the quality of implementation, the goals of the program's implementation should and should be stated clearly. As currently written, the introduction section seems to simply say that the program was mandated and that various aspects of the program were examined. Page 8, paragraph 3, methods section: A preposition seems absent. sentence should read "we agreed upon research governance and reporting strategies". Page 8, paragraph four, "methods to address objectives": The first sentence describes "systematically mapping information supplied by providers". It's not clear to me what this means, and I suspect that the investigators used written materials as the input. Page 13, paragraph one: I don't think the "details of themes" is detailed enough. For example, what does "social factors, such as group support" mean? There is one example of a quote in the theme of group support in Box 2. Should the reader interpret this as meaning most or all of the comments on this topic were positive? Page 22, paragraph three and page 23, paragraph 1: these two paragraphs are informative but don't seem to be necessary to the discussion section. I would prefer that the discussion section instead interprets the findings and talks more about how these findings will affect for their implementation efforts.
--

VERSION 1 – AUTHOR RESPONSE

Reviewer: 1

Reviewer Name: Alvaro Sánchez Pérez

Institution and Country: Primary Care Research Unit of Bizkaia-Osakidetza, SPAIN

Competing Interests: None declared

This is a very relevant manuscript for those involved in the scale-up of proven clinical interventions. We thank Alvaro Sánchez Pérez for this appreciation of our work

Specifically, the objective of the study is to evaluate the demonstrator phase and first wave roll-out of the NHS diabetes prevention program (DPP) in England, to inform subsequent phase. To do so, they have examined: (i) intervention design and provision, in relation to the evidence base, and procedures to assess intervention fidelity; (ii) risk assessment procedures and recruitment pathways and (iii) data collection, monitoring and evaluation.

The evaluation methods included structured document review, comparing the program with the recommendations in NICE PH38, and qualitative research involving individual interviews and interactive focus group discussions with key stakeholders, including service users

Due to the relevance at public health level of the general NHS Diabetes Prevention Program in England and the intended goals of the present study (informing the rule-out of future phases in a more efficient way, for both implementation and evaluation), the present manuscript has several core elements in which it might be improved.

Major points

First, it is not clear what and how within the MRC framework the present evaluation fits. What is the specific contribution or the specific question that the study is trying to answer within such framework? Please comment on this, especially, when the study seems to be the Phase IV translation stage.

Response: We thank the reviewer for this suggestion, which has helped to clarify an important goal of our study.

We have clarified the specific contribution in terms of providing recommendations based on findings to inform decision makers (in both the abstract and the methods). This is in line with the implementation phase requirements for evaluations, as described in the MRC 2008 framework for development and evaluation of complex interventions to improve health.

These changes are tracked in the document and copied below with changes highlighted for convenience.

Evaluation of the demonstrator phase and first wave roll-out of the NHS diabetes prevention programme (DPP) in England, to inform subsequent phases. To examine: (i) intervention design, provision and fidelity assessment procedures; (ii) risk assessment and recruitment pathways and (iii) data collection for monitoring and evaluation and to provide recommendations based on findings to inform decision makers on programme quality, improvements and future evaluation

In planning the evaluation we drew on the Medical Research Council (MRC) guidance for development and evaluation of complex interventions to improve health¹² and the MRC guidance for process evaluation of complex interventions to improve health.¹³

Comment: Second, and linked to the previous point, the present evaluation only considers the clinical intervention level, not informing about the most important aspect in my opinion when addressing the transfer/dissemination/scale-up of clinical interventions: the implementation strategy used to enhance the adoption and execution of the clinical intervention in the studied sites/contexts. Specially, when it is pretty likely to be variable among sites and contexts, raising issues related to fidelity-flexibility, not only on how the clinical intervention has been organized and carried out, but on the implementation strategies used.

Within the different types of program evaluation (process, outcome, impact, etc.), the study actually describes factors (organized by subject) that have facilitated / prevented the 3 aspects of evaluation related to clinical intervention based on review of documentation and comparison to recommended program, and key agent opinions and perceptions extracted from interviews. As a consequence, the true contribution of the study, taking into account its strengths and weaknesses, should be more clearly specified in both the title and the body of the manuscript.

Response: We thank the reviewer for drawing our attention to this need for clarification of the scope of our evaluation. We have suggested an addition to the title that might clarify. However, if the reviewer thinks that this still doesn't quite hit the spot, and they are prepared to offer a better suggestion we would be appreciative.

Our evaluation was conducted at an early stage in the roll out of the NHS DPP. The scope of our evaluation was to focus on the nationally commissioned programme and implementation strategies, and support for its future evaluation, rather than on details of site-specific organisational strategies to facilitate uptake of the programme in different localities. We made important, specific recommendations to the NHS DPP management group that were based on our findings, (such as to improve recruitment data collection - please see highlight in the manuscript). These recommendations, if actioned by the management, would support the assessment of implementation strategy in future evaluations.

We also reported on the early views of local commissioners in relation to their experiences of implementation, but we were aware that these views were recorded at an early stage in the process. We have added to the discussion of strengths and limitations of our study, (p22) as suggested, to clarify the focus of this early phase study and the need to further consider local implementation strategies as the programme is embedded and rolled out across the country.

The comparison of how the programs were organized / implemented with respect to the recommendations is limited, describing in a table the operational organization of the program according to 5 sites and explaining some discrepancies in the use of BCTs and fidelity aspects. An exhaustive or at least better organized description of how the clinical intervention program has been organized / implemented at each site, and its deviation from what is recommended, would help, especially those different ways of putting it into practice. But also, it would be very interesting to know which of these procedures are successful or not, based on not only perceptions of the agents but on numbers (quantitatively). The NICE baseline measurement tool can be a possible guide on how to structure a table for that description.

We have provided a table comparing intervention components across first wave provider organisations as appendix A (supplementary file). Detailed evaluation of the ways in which first wave sites operationalise the programme was beyond the scope of this early phase evaluation, which was designed to inform the national programme strategy and future implementation and evaluations. In our reports we made recommendations on the need for further evaluation of site specific operational variation, with specific details of data requirements (eg. the collection of data to provide detail on routes to recruitment) to facilitate such evaluations.

We agree that quantitative data would be of interest. However, these data were not available to us for this early stage evaluation. Part of our remit was to make recommendations on data collection that would support future evaluation. We made data collection recommendations, but action on these recommendations was a management decision.

Comment: Though it is a "qualitative" evaluation, and not a formal program process evaluation, further quantitative information will help to understand the magnitude of the study (ie., reach of the different screening procedures, intervention delivery procedures...by each site). This information will also help to describe which procedures and strategies are more successful.

Response: As above the lack of quantitative data at this early stage (other than the collated management report data, which is included in our paper) is regrettable. We hope that the data collection requests that we made will ensure these data are suitable and available for further evaluation at later times. We have added to the strengths and limitations to position the recommendations within the MRC framework. We have also explained the value of eliciting formal response to our recommendations, from the management group, to facilitate impact evaluation, which would be of implementation interest. (page 22)

Comment: An implementation framework or model that guides the design or at least the analysis, synthesis, and interpretation of these determinants of implementation (ie, CFIR framework) is missed. This would help not only to understand which determinants are key and why, but also to identify and / or design strategies to address such determinants in future program roll-out.

Response: We thank the reviewer for suggesting the CFIR framework. We have not used this as part of our evaluation as the focus at this early stage was on evaluation of the national programme and roll-out procedures rather than local operational strategies for the implementation of this programme. The local site-specific implementation of a nationally commissioned and funded programme, which is delivered by four dedicated provider organisations is an interesting implementation model. We do have a rich set of qualitative data, but there is a concern that these data are collected very early in the roll-out.

Comment: Linked to this previous point, and within the parts describing the framework and methods that have guided the evaluation, the authors refer to the mapping of the retrieved information from providers against the NICE recommendations, equality indicators, fidelity, data collection and quality assurance procedures. But none of these matched or mapped elements are clearly displayed or presented within the manuscript.

Response: The mapping for first wave programmes used the recommendations in NICE guidance and PROGRESS indicators. The NHS DPP national service specification and the baseline provider documents that the provider organisations submitted as part of their procurement process were mapped against these. These provider organisations are commercial enterprises and as such these detailed documents were made available to the evaluation team with confidentiality understanding in place. We are able to publish summaries of the findings from this mapping procedure. We are not able to make these mapping tables open access at this time, due commercial interests and confidentiality arrangements, which were agreed with the management group. However, if a research team were to request access to these data (tables) for specific research use we would enquire if these might be made available for defined research purposes.

Comment: They also refer to the development of some tools (page 9, last paragraph) for use in intervention feasibility, acceptability and fidelity assessment. An example of these tools, due to the help that may offer to the reading public, is highly recommendable.

Response: At present these tools are in a draft stage and as such they are untested (other than through limited PPI processes). We hope to be able to assess the utility of the tools and then, if they can be tested and found to be fit for purpose, we would be pleased to then make them available more generally. In the meantime we would be pleased to hear from others who are involved with interventions for prevention of type 2 diabetes in people at high risk and who would like to work with us to assess the utility of these tools.

Minor points

Abstract-findings: more precise or concrete results summarizing actual obtained results
We have added to the abstract to clarify as suggested.

Comment: A Figure Flow-diagram is highly recommendable to fully describe the structure and process (phases, sites, centers, professionals, patients referred, etc) of the study.

Response: We have supplied a timeline as a supplementary file.

Reviewer: 2

Reviewer Name: Neal Barshes

Institution and Country: Baylor College of Medicine, Houston, Texas, United States of America

Competing Interests: No competing interests

Page 8, paragraph 1, Introduction section: what was the goal of this study? This should be clearly stated clearly. This need not to be a hypothesis to prove or disprove, but if it is a study to ensure the quality of implementation, the goals of the program's implementation should and should be stated clearly. As currently written, the introduction section seems to simply say that the program was mandated and that various aspects of the program were examined.

Response: We thank the reviewer for highlighting this need for clarification and have responded accordingly to clarify the goal as recommendations to the management group to inform decision makers on programme quality, improvements and future evaluation.

Page 8, paragraph 3, methods section: A preposition seems absent. sentence should read "we agreed upon research governance and reporting strategies".

Response: Thank you, we have rephrased.

Page 8, paragraph four, "methods to address objectives": The first sentence describes "systematically mapping information supplied by providers". It's not clear to me what this means, and I suspect that the investigators used written materials as the input.

Response: Thank you, we did use written materials and we have added clarification.

Page 13, paragraph one: I don't think the "details of themes" is detailed enough. For example, what does "social factors, such as group support" mean? There is one example of a quote in the theme of group support in Box 2. Should the reader interpret this as meaning most or all of the comments on this topic were positive?

Response: We agree the details are limited and we have added some further explanation. For this paper we aimed to provide a summary of the entire evaluation within a reasonable word count. This was challenging and inevitably some of the informative detail is sparse to keep within this word limit. We do have a rich qualitative data set and there may be opportunities to revisit and expand on this aspect of our evaluation in the future.

Page 22, paragraph three and page 23, paragraph 1: these two paragraphs are informative but don't seem to be necessary to the discussion section. I would prefer that the discussion section instead interprets the findings and talks more about how these findings will affect for their implementation efforts.

Page 22 paragraph 3 and page 23 paragraph 1: we have added to these paragraphs to clarify why we included this information.

Response: In response to the request for interpretation of findings we have added consideration of the utility of including (i) specific behavioural change theory in the logic model in comparison to listing and detailing (ii) specific BCTs with their associated delivery strategies in the logic model. This is especially relevant in support of future evaluation procedures.

VERSION 2 – REVIEW

REVIEWER	Alvaro Sánchez Primary Care Research Unit of Bizkaia, Basque Health Service- OSAKIDETZA. SPAIN
REVIEW RETURNED	09-Nov-2017

GENERAL COMMENTS	Authors have correctly answered the issues raised by the reviewers or at least, as good as possible due to some limitations of the performed study (eg., they do not count with quantitative data). The scientific contribution is not cutting edge. However the manuscript has some interesting lessons for those involved in the translation of diabetes prevention programs.
---

REVIEWER	Neal Barshes Baylor College of Medicine United States of America
REVIEW RETURNED	02-Nov-2017

GENERAL COMMENTS	Adequate responses to critiques.
----------------------------------